# Association of *miR-149* T>C and *miR-196a2* C>T Polymorphisms with Colorectal Cancer Susceptibility: A Case-Control Study

**DOI:** 10.3390/biomedicines11092341

**Published:** 2023-08-23

**Authors:** Bayram Bayramov, Nuru Bayramov, Hazi Aslanov, Nigar Karimova, Karim Gasimov, Ilham Shahmuradov, Christoph Reißfelder, Vugar Yagublu

**Affiliations:** 1Laboratory of Human Genetics, Genetic Resources Institute of Ministry of Science and Education, Baku AZ1106, Azerbaijan; dr.bayrambayramov@gmail.com (B.B.); nigar.karimova.23@gmail.com (N.K.); 2Department of Surgery, Azerbaijan Medical University, Baku AZ1022, Azerbaijan; nurubay2006@yahoo.com; 3Department of Surgery, Scientific Center of Surgery, Baku AZ1122, Azerbaijan; haslanov@yahoo.com; 4Laboratory of Molecular and Cellular Biochemistry, Institute of Biophysics of Ministry of Science and Education, Baku AZ1141, Azerbaijan; gasimovk@yahoo.com; 5Bioinformatics Lab, Institute of Molecular Biology and Biotechnologies of Ministry of Science and Education, Baku AZ1141, Azerbaijan; ilhambaku@gmail.com; 6Integrative Biology Lab, Institute of Biophysics of Ministry of Science and Education, Baku AZ1141, Azerbaijan; 7Department of Surgery, Universitätsmedizin Mannheim, Medical Faculty Mannheim, Heidelberg University, 68167 Mannheim, Germany; christoph.reissfelder@umm.de

**Keywords:** colorectal cancer, polymorphism, non-coding RNA, microRNA, PCR-RFLP

## Abstract

The principal aim of the current study was to investigate the relationship between *miR-149* T>C (rs2292832) and *miR-196a2* C>T (rs11614913) small non-coding RNA polymorphisms and the risk of developing CRC in the Azerbaijani population. The study included 120 patients diagnosed with CRC and 125 healthy individuals. Peripheral blood samples were collected from all the subjects in EDTA tubes and DNA extraction was performed by salting out. Polymorphisms were determined using the polymerase chain reaction-restriction fragment length polymorphism (PCR-RFLP) method. While comparing without gender distinction no statistical correlation was found between the heterozygous TC (OR = 0.66; 95% CI = 0.37–1.15; *p* = 0.142), mutant CC (OR = 1.23; 95% CI = 0.62–2.45; *p* = 0.550), and mutant C (OR = 1.03; 95% CI = 0.72–1.49; *p* = 0.859) alleles of the *miR-149* gene and the CT (OR = 1.23; 95% CI = 0.69–2.20; *p* = 0.485), mutant TT (OR = 1.29; 95% CI = 0.67–2.47; *p* = 0.452), and mutant T (OR = 1.17; 95% CI = 0.82–1.67; *p* = 0.388) alleles of the *miR-196a2* gene and the risk of CRC. However, among women, *miR-149* TC (OR = 0.43; 95% CI = 0.19–1.01; *p* = 0.048) correlated with a reduced risk of CRC, whereas *miR-196a2* CT (OR = 2.77; 95% CI = 1.13–6.79; *p* = 0.025) correlated with an increased risk of CRC. Our findings indicated that *miR-149* T>C (rs2292832) might play a protective role in the development of CRC in female patients, whereas the *miR-196a2* (rs11614913) polymorphism is associated with an increased risk of CRC in women in the Azerbaijani population, highlighting the importance of gender dimorphism in cancer etiology.

## 1. Introduction

Colorectal cancer (CRC) stands as one of the prevalent forms of cancer and poses a significant global health concern. It is a multifactorial disease observed in both men and women. The number of new cases including mortality has increased in recent years [1,2]. Depending on the socio-economic development of the countries, it is estimated that the total number of deaths from rectal and colon cancer will increase substantially by 2035 [3,4]. It has been determined that the risk of morbidity and mortality is significantly reduced in countries where invasive and noninvasive screening programs are carried out [5,6]. Chromosome instability (CIN, 80–85% of all CRC cases), microsatellite instability (MSI) including mismatch repair genes (MMR), and CpG island methylator phenotype (CIMP), which provides silencing of oncogenes and tumor suppressor genes by hypermethylation, constitute the molecular basis of the disease [7]. Besides DNA promoter methylation, other epigenetic modifications like non-coding RNAs and histone modifications have also been associated with various disorders including CRC [8]. Short (~20–25 nt) single-stranded microRNAs (miRNAs), a type of non-coding RNA, regulate gene expression at the post-transcriptional level by binding (either degrading them or reducing their expression) to the 3′ untranslated region (3′UTR) of mRNA [9,10]. Although miRNAs possess tumor suppressor and oncogene functions in various cancers and are important in carcinogenesis, they also contribute to different cellular biological processes such as cell growth, differentiation, and cell division [11]. Single-nucleotide polymorphisms (SNPs) in genes encoding miRNAs can adversely affect mature miRNA formation and interaction with target mRNAs, thereby affecting all stages of miRNA biogenesis [12]. In recent studies, SNPs associated with miRNAs in various diseases have been reported and recommended as candidate genetic biomarkers [13,14]. SNPs in *miR-149* and *miR-196a2* genes investigated in different countries and ethnic groups have been reported to be associated with colorectal cancer [15], lung cancer [16], breast cancer [17], hepatocellular carcinoma [18], gastric cancer [19], cardiovascular diseases [20], etc. Since the *miR-149* and *miR-196a2* gene polymorphisms have never been studied in CRC patients in our population, consistent data is unavailable. In this case–control study, we have investigated the relationship between *miR-149* and *miR-196a2* microRNA gene polymorphisms and subjects’ age, gender, clinicopathological parameters, smoking and alcohol use parameters, and CRC susceptibility for the first time in the Azerbaijani population.

## 2. Materials and Methods

### 2.1. Subjects

The study included 120 patients diagnosed with CRC at the Scientific Surgery Center of Azerbaijan and the Educational-Surgical Clinic of Azerbaijan Medical University during the period 2018–2021. The study protocol was approved by the Ethics Committee of Genetic Resources Institute and written informed consent was obtained from each patient. Sporadic colorectal cancer cases were included in this study and a history of inflammatory bowel diseases and hereditary colon cancer syndromes were excluded. The histopathology data of the tumors (tumor grade and stage) were specified in the pathology report. We included 125 healthy individuals without a family history of cancer who applied for a routine colonoscopy. After obtaining their consent, additional information such as age, smoking status, and alcohol consumption from both patients and controls were recorded. Peripheral venous blood was collected from each individual in a tube with EDTA for genomic DNA extraction. DNA extraction was carried out using the salting out technique in the Laboratory of Human Genetics of the Genetic Resources Institute, and DNA samples were stored at −20 °C until they were used in the next step. Quantitative and qualitative parameters of DNA were measured in NanoDrop™ 2000/2000c spectrophotometer (Thermo Fisher Scientific, Waltham, MA, USA). 

### 2.2. Genotyping

Genotypes of *miR-149* T>C (rs2292832) and *miR-196a2* C>T (rs11614913) genes were determined on agarose gel using PCR-RFLP method. PCR reactions of the studied genes in a volume of 25 µL contained the following components: 2.5 µL 10×PCR buffer, 2.5 µL MgCl_2_ (50 mM), 0.25 µL dNTP mixture (20 mM), 0.5 µL (10 pmol/µL) from each of the forward and reverse primers, 0.25 µL (5 U/µL) Tag polymerase (Solis BioDyne, Tartu, Estonia), 2 µL genomic DNA (50 ng/µL) and 16.5 µL distilled water (dH_2_O). Amplification conditions for PCR reactions (Applied Biosystems, Waltham, CA, USA) involved initial denaturation at 95 °C for 5 min, 35 cycles at 95 °C for 30 s, annealing for *miR-149* T>C gene at 61 °C for 45 s, annealing for *miR-196a2* C>T gene at 59 °C for 45 s and 2 min at 72 °C, followed by 5 min final elongation at 72 °C. 

After electrophoresis of PCR products in 1.5% agarose gel, amplicons were purified according to the QIAquick PCR Purification Kit (Qiagen, Hilden, Germany) protocol. Subsequently, *PvuII* (New England Biolabs, Massachusetts, USA) for *miR-149* T>C and *MspI* (New England Biolabs, Ipswich, MA, USA) restriction enzymes for *miR-196a2* C>T polymorphism were used to determine genotypes.

Restriction fragments were visualized on 2% agarose gel stained with Ethidium Bromide under UV gel documentation system (Figure 1 and Figure 2. The *miR-149* T>C genotypes were observed using agarose gel electrophoresis. The TT genotype appeared as a single band at 254 bp, indicating a homozygous uncut pattern. The TC genotype appeared as three bands at 254 bp, 196 bp, and 60 bp, representing a heterozygous pattern. The CC genotype appeared as two bands at 196 bp and 60 bp, indicating homozygous mutant genotypes. The different genotypes of *miR-196a2* C>T were observed using agarose gel electrophoresis. The homozygous CC genotype appeared as a single band at 149 bp, the heterozygous CT genotype appeared as two bands at 149 bp and 125 bp, and the homozygous mutant TT genotype appeared as a single band at 125 bp. The sequence of primers, amplicon size, and restriction enzymes and fragments used in the study are listed in Table 1. 

An amount of 10% of all the samples were randomly selected for validation purposes. The obtained results were genotyped repeatedly.

### 2.3. Statistical Analysis

The biostatistical analysis of the results was performed using the SPSS package (ver. 22, SPSS, Chicago, IL, USA). The association between the parameters was evaluated by Pearson’s chi-square test (χ^2^) and Fisher’s exact test. Fisher’s exact test for contingency tables more than 2 × 2 was performed with Social Science Statistics (http://www.socscistatistics.com/tests/chisquare2/Default2.aspx; accessed on 10 October 2022). A binary logistic regression was carried out to calculate the odds ratios (ORs) with 95% confidence intervals (CIs). All the statistical tests were two-sided; the significance level was taken *p* < 0.05.

## 3. Results

This case–control study included 120 patients with CRC and 125 healthy individuals. Table 2 presents the demographic parameters of the patient and control groups. Of the cancer patients, 68 (56.7%) were males and 52 (43.3%) were females, whereas 55 (44%) of the control group were males and 70 (56%) were females. The age range of the patients was 35–84 and 32–82 in the control group, while the mean age was 63 ± 10.1 and 60.9 ± 11.5, respectively. According to the pathology reports, grades of tumors were evaluated as 11.7% for G1, 56.7% for G2, 27.5% for G3, and 4.2% for G4, respectively. On the other hand, tumor stages were determined as 11.7% for T1, 15.8% for T2, 63.3% for T3, and 9.2% for T4, respectively. There was a lack of association among patients’ tumor grade and stage and *miR-149* and *miR-196a2* polymorphisms. Age, gender, and smoking and alcohol consumption data of patients and controls were compared, and no statistical correlation was observed (*p* > 0.05).

Table 3 describes the genotype and allele frequencies for the single-nucleotide polymorphisms of the *miR-149* T>C and *miR-196a2* C>T genes. The wild-type TT genotype of the *miR-149* gene was found in 44.2% and 39.2%, heterozygous TC in 32.5% and 44%, and homozygous mutant CC genotype in 23.3% and 16.8% of the patients and control groups, respectively. A statistical correlation was not observed between the risk of CRC and both heterozygous TC (OR = 0.66; 95% CI = 0.37–1.15; *p* = 0.142) and mutant CC (OR = 1.23; 95% CI = 0.62–2.45; *p* = 0.550) genotypes of the *miR-149* gene. The frequency of the T allele was 60.4% in patients and 61.2% in healthy individuals. The mutant C allele was relatively higher in the patients (39.6%) as compared to the control group (38.8%). There were no statistically significant differences between the two groups in terms of mutant C (OR = 1.03; 95% CI = 0.72–1.49; *p* = 0.859) alleles. Furthermore, when the *miR-196a2* C>T polymorphism was evaluated, the heterozygous CT (OR = 1.23; 95% CI = 0.69–2.20; *p* = 0.485) and homozygous mutant TT genotype (OR = 1.29; 95% CI = 0.67–2.47; *p* = 0.452) were found more frequently (40.8% and 26.7%) in the CRC patients as compared to the control group. The incidence of the C allele was 52.9% in the patients and 56.8% in the control group. Additionally, the mutant T allele was detected in 47.1% of the patients and in 43.2% of the control group. However, there was no statistically significant association between the *miR-196a2* C>T polymorphism and CRC risk. 

The polymorphisms *miR-149* T>C and *miR-196a2* C>T were compared in the subject groups according to gender and age (Table 4 and Table 5). The *miR-149* TC genotype was higher in healthy men (38.2%) and the homozygous mutant CC was found more frequently in male patients (19.1%). In addition, no statistical correlation was observed between the *miR-149* T>C polymorphism and the risk of CRC when the males in the study groups were compared (*p* > 0.05). Conversely, when female patients were compared with healthy women, a statistical association was found between the *miR-149* TC genotype and reduced CRC risk (OR = 0.43; 95% CI = 0.19–1.01; *p* = 0.048). However, the *miR-149* CC genotype was more frequent in female patients, but no statistical difference was found. Due to the comparison, the frequency of the mutant CC genotype was higher in patients under 60 years old compared to patients over 60 (45.8% and 43.1%). However, no statistical difference was observed between *miR-149* genotypes and disease risk according to age.

Similarly, the distribution of the *miR-196a2* C>T polymorphism by age and gender is presented in Table 4. Although the mutant TT genotype (23.5%) was predominant in male patients as compared to healthy men, it did not indicate statistical significance (OR = 0.99; 95% CI = 0.38–2.56; *p* = 0.979). Both heterozygous CT (48.1%) and homozygous mutant TT (30.8%) genotypes were more common in female patients than in healthy women. Particularly, the heterozygous CT genotype was associated with an increased risk of CRC (OR = 2.77; 95% CI = 1.13–6.79; *p* = 0.025). No statistical difference was observed when comparing *miR-196a2* C>T polymorphism based on age in patients and control groups. However, CT and TT genotypes were more common in patients over 60 years old.

We calculated the distribution of genotypes with respect to the smoking and alcohol statuses of the patients (Table 6). Furthermore, both the heterozygous TC (31.4%) and mutant CC (28.6%) genotypes of the *miR-149* gene were higher in smokers, while TC and CC genotypes were more common in non-drinkers. Similarly, the polymorphism of the *miR-196a2* C>T gene was analyzed, wherein the heterozygous CT was higher in non-smokers (42.9%) and non-drinkers (43.8%), while mutant TT was more common in smokers (28.6%) and alcohol drinkers (31.3%). However, when the polymorphisms of both genes were compared between the subject groups, no statistical difference was observed (*p* > 0.05).

Table 7 presents the distribution of *miR-149* T>C genotypes based on tumor stage and grade. The mutant CC genotype was found to be more prevalent among patients with tumor grade G3. When it comes to tumor stages, the TC genotype was more frequently observed in T3 cases, while the CC genotype was more prevalent in T1 cases. However, there were no significant statistical differences in genotype distribution concerning tumor grade and stages (*p* > 0.05).

Moreover, the heterozygote CT and mutant TT of *miR-196a2* C>T was higher in tumor grades G1 and G4, respectively (Table 8). As for tumor stages, the CT and TT genotypes were more frequent in T2 stages. It is important to mention that the correlation between the distribution of genotypes and tumor stages was statistically significant (*p* < 0.05).

## 4. Discussion

The application of screening programs is important in clinical diagnosis for the early and timely detection of precancerous pathologies and malignant tumors. The recent widespread use of noninvasive molecular-based analyses has provided essential opportunities for early detection, diagnosis, metastasis, understanding of drug resistance mechanisms, and personalized medicine [21]. SNPs in the miRNA encoding sequence can directly affect the biogenesis of miRNA, thereby influencing the transcription of pri-miRNAs in the nucleus, the formation of mature miRNA, and the interaction of miRNAs with target mRNA [22]. In our study, we examined the relationship between the *miR-149* T>C and *miR-196a2* C>T gene polymorphisms, the most widely studied in miRNA genes, and CRC risk for the first time in the Azerbaijani population. 

We found no statistical correlation between the genotype and allele frequency of the *miR-149* rs2292832 and *miR-196a2* rs11614913 and the risk of CRC while comparing without gender distinction. Similarly, Hezova et al. reported that *miR-196-a2, miR-27a*, and *miR-146a* gene polymorphisms were not associated with CRC risk in the Czech population [23]. A meta-analysis showed that lung, breast, and colorectal cancers were not associated with the *miR-149* polymorphism in a large study group [24]. Furthermore, the gene polymorphism of *miR-196a2* was not associated with CRC susceptibility in a meta-analysis of European studies [25]. The rs11614913 polymorphism of the *miR-196a2* gene was not found to be associated with CRC in the Greek population either [26]. In the meta-analysis performed in China, no significant correlation was observed in the overall results for rs2292832 [27]. A study conducted in Iran showed that rs2292832 of the *miR-149* gene and rs11614913 polymorphisms of the *miR-196a2* gene were not associated with the risk of CRC [28,29]. 

Another meta-analysis found no association between the *miR-146* and *miR-149* polymorphisms and CRC, whereas SNP in the *miR-196* gene was associated with CRC in the Asian population [30]. Moreover, Choupani et al. stated that the rs11614913 polymorphism of the *miR-196a2* gene is associated with the risk of CRC only in Asia and not in the Caucasus. It has also been reported that rs2292832 of the *miR-149* gene does not affect cancer in the general population, but the recessive model increases the risk of CRC [31]. However, in our study, there was no relationship between the dominant and recessive model and the risk of disease. 

In this study, the tumor grade and stage were not associated with the *miR-149* T>C polymorphism. Our results are in concordance with recent studies [15]. Concerning the *miR-196a2* C>T polymorphism, no association was observed between tumor grade and the polymorphism. However, a statistically significant correlation was identified between the polymorphism and tumor stage. Likewise, Zhu and colleagues observed a significant association between the *miR-196a2* C>T variant and the susceptibility of patients with advanced-stage tumors (Dukes C and D) [20].

In a study by Wang, the *miR-196a2* rs11614913 polymorphism was not associated with the pathological parameters of the tumor [32]. However, there is a statistically significant relationship between the SNP and the stage of the tumor. In contrast, Chen et al. reported that either genotype or allele frequencies of the *miR-196a2* gene do not contribute toward the risk of CRC, demonstrating no statistically significant relationship between *miR-196a2* rs11614913 and tumor grade, stage, tumor invasion, and lymph node metastasis status [33]. 

However, when gender distinction is considered in our study, rs2292832 of *miR-149* is associated with a lower risk of CRC among females and may play a protective role against developing CRC in women. Our work highlights the importance of sex dimorphism in cancer once more. There is a growing body of recent evidence supporting the notion of sexual dimorphism in cancer, which was introduced in 2016 [34] and is based on differences in tumor biology between tumors arising in males and females. It is well known that CRC incidence rates are clearly sexually dimorphic in every region of the world [35], with female incidences being lower than males. On the other hand, according to several retrospective studies, women with CRC have a higher survival rate than men [36]. For instance, women in a German cohort study of 185,967 patients had significantly higher survival rates than men [37]. Therefore, it is important to bring more insight into the genetic architecture of colon cancer by taking into account gender differences. 

In order to clarify whether other studies showed similar results when gender is considered, we conducted a literature search. Ranjbar et al. found a significant relationship between the *miR-149* rs2292832 polymorphism and gender/age in the Iranian population [28]. The meta-analysis investigating digestive system cancers found no relationship between the *miR-149* rs2292832 T/C polymorphism and ethnicity and smoking, including sex [38]. Zhang et al. did not find a relationship between *hsa-miR-605* (rs2043556) and *hsa-miR-149* (rs2292832) SNPs in women with CRC and gastric cancer; however, researchers reported that these polymorphisms showed protective effects in men [39]. In addition, the rs2292832 polymorphism in *miR-149* was found to be associated with a reduced cancer risk in other cancer diseases such as breast cancer [40] and cervical cancer [41]. 

Moreover, we found *miR-196a2* heterozygous CT to be associated with a higher risk of CRC when comparing female patients with healthy subjects. In a study conducted in the Indian population, the rs11614913 CT genotype increased the risk of breast cancer in women [17]. Wang et al. reported that the *miR-196a2* rs11614913 polymorphism was also associated with breast cancer susceptibility in women in Chinese and Indian populations [42]. In contrast, rs11614913 is associated with decreased risk of esophageal squamous cell carcinoma in female patients and patients who never smoke or drink [43].

## 5. Conclusions

We investigated a possible association between the *miR-149* gene T>C (rs2292832) and *miR-196a2* gene C>T (rs11614913) polymorphisms and the risk of CRC in an Azerbaijani population. Our results have demonstrated that the *miR-149* T>C (rs2292832) heterozygous TC genotype was associated with a lower risk of CRC among females and might play a protective role against the development of CRC in women. On the other hand, *miR-196a2* heterozygous CT in our study has been shown to be associated with a higher risk of CRC in female patients. 

This study is an attempt to provide preliminary evidence that pretreatment genotyping of *miR-149* rs2292832 and *miR-196a2* polymorphisms can help predict the development of CRC. The present study is the first to report *miR-149* T>C and *miR-196a2* C>T polymorphisms as risk factors for CRC in an Azerbaijani population. However, we should also note that our research has certain limitations. Firstly, we had only a small sample size of females included in our study, and so we may need to increase the sample size to clearly demonstrate the effect of heterozygous genotypes of both *miR-149* T>C and *miR-196a2* C>T polymorphisms on CRC risk. In addition, the participants were from only two centers, the Scientific Surgery Center of Azerbaijan and the Educational-Surgical Clinic of Azerbaijan Medical University. The extent to which the present findings can be applied broadly should be cautiously approached, as no validation study has been undertaken with a distinct Azerbaijani population or a population of a different ethnicity. Furthermore, in our study, we did not investigate how polymorphisms affect gene expression, and it is possible that these genes could be analyzed in detail using next-generation sequencing systems, and their effect on CRC risk could be demonstrated.

In conclusion, our findings suggest that *miR-149* rs2292832 and *miR-196a2* polymorphisms may have a role in the genetic etiology of CRC in women in the Azerbaijani population.

## Figures and Tables

**Figure 1 biomedicines-11-02341-f001:**
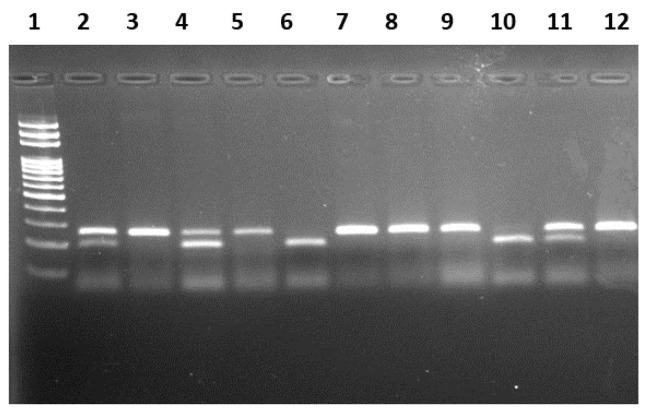
Genotypes of *miR-149* gene polymorphism determined by PCR-RFLP methods in agarose gel. DNA Ladder (100 bp): Lane-1. Wild-type TT: Lane-3, 5, 7, 8, 9, 12. Heterozygous TC: Lane-2, 4, 11. Homozygous mutant CC: Lane-6, 10.

**Figure 2 biomedicines-11-02341-f002:**
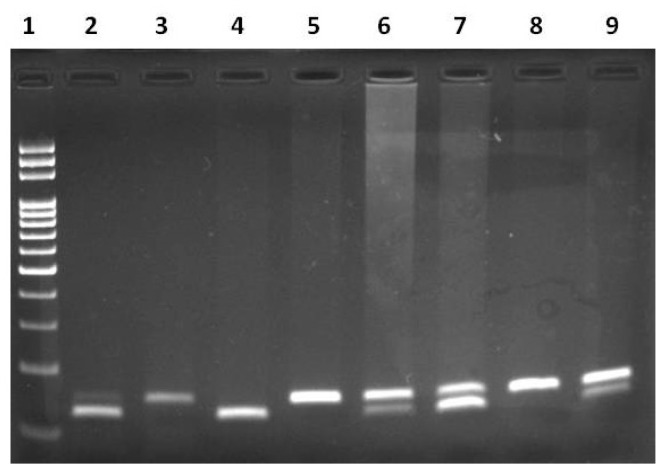
Genotypes of *miR-196a2* gene polymorphism determined by PCR-RFLP methods in agarose gel. DNA Ladder (100 bp): Lane-1. Wild-type CC: Lane-3, 5, 8. Heterozygous CT: Lane-6, 7, 9. Homozygous mutant TT: Lane-2, 4.

**Table 1 biomedicines-11-02341-t001:** MiRNA SNPs, specific primer sequences, and restriction enzymes.

Polymorphisms	Sequence of Primers	PCR Products	Restriction Enzymes	Restriction Fragments
*miR-149* T>C (rs2292832)	F: TGTCTTCACTCCCGTGCTTGTCC R: TGAGGCCCGAAACACCCGTA	254 bp	*PvuII*	T allele: 254 bp C allele: 196 bp + 60 bp
*miR-196a2* C>T (rs11614913)	F: CCCCTTCCCTTCTCCTCCAGATA R: CGAAAACCGACTGATGTAACTCCG	149 bp	*MspI*	C allele: 149 bp T allele: 125 bp

**Table 2 biomedicines-11-02341-t002:** Demographic and clinical parameters related to cases and controls.

Characteristics	Patients N = 120 (%)	Controls N = 125 (%)	*p*-Values
Gender			
Male	68 (56.7%)	55 (44%)	0.259
Female	52 (43.3%)	70 (56%)	
Age			
Age interval	35–84	32–82	0.152
Mean	63 ± 10.1	60.9 ± 11.5	
Histological Grade			
G1	14 (11.7%)		
G2	68 (56.7%)		
G3	33 (27.5%)		
G4	5 (4.2%)		
Tumor Stage			
T1	14 (11.7%)		
T2	19 (15.8%)		
T3	76 (63.3%)		
T4	11 (9.2%)		
Smoking Status			
Smokers	35 (29.2%)	58 (46.4%)	0.554
Non-Smokers	77 (64.2%)	61 (48.8%)	
Unknown	8 (6.6%)	6 (4.8%)	
Alcohol consumption			
Yes	32 (26.7%)	41 (32.8%)	0.612
No	80 (66.7%)	79 (63.2%)	
Unknown	8 (6.6%)	5 (4%)	

**Table 3 biomedicines-11-02341-t003:** Distribution of genotypes and allele frequencies of *miR-149* and *miR-196a2* genes in study groups.

*miR-149* T>C	Cases N = 120 (%)	Controls N = 125 (%)	OR (95% CI)	*p*-Values
Genotype				
TT	53 (44.2)	49 (39.2)	1	-
TC	39 (32.5)	55 (44)	0.66 (0.37–1.15)	0.142
CC	28 (23.3)	21 (16.8)	1.23 (0.62–2.45)	0.55
Dominant model				
TT	53 (44.2)	49 (39.2)	1	-
TC+CC	67 (55.8)	76 (60.8)	1.23 (0.74–2.04)	0.43
Recessive model				
TT+TC	92 (76.7)	104 (83.2)	1	-
CC	28 (23.3)	21 (16.8)	0.66 (0.35–1.25)	0.201
Allele				
T	145 (60.4)	153 (61.2)	1	-
C	95 (39.6)	97 (38.8)	1.03 (0.72–1.49)	0.859
*miR-196a2* C>T	Cases	Controls	OR (95% CI)	*p*-Values
	N = 120 (%)	N = 125 (%)		
Genotype				
CC	39 (32.5)	47 (37.6)	1	-
CT	49 (40.8)	48 (38.4)	1.23 (0.69–2.20)	0.485
TT	32 (26.7)	30 (24)	1.29 (0.67–2.47)	0.452
Dominant model				
CC	39 (32.5)	47 (37.6)	1	-
CT+TT	81 (67.5)	78 (62.4)	1.25 (0.74–2.12)	0.403
Recessive model				
CC+CT	88 (73.3)	95 (76)	1	-
TT	32 (26.7)	30 (24)	1.52 (0.65–2.05)	0.631
Allele				
C	127 (52.9)	142 (56.8)	1	-
T	113 (47.1)	108 (43.2)	1.17 (0.82–1.67)	0.388

**Table 4 biomedicines-11-02341-t004:** Distribution of *miR-149* genotypes in terms of age and gender.

Genotypes	Cases N = 68 (%)	Controls N = 55 (%)	OR (95% CI)	*p*-Values
Males				
TT	30 (44.1)	25 (45.5)	1	-
TC	25 (36.8)	21 (38.2)	0.99 (0.45–2.18)	0.984
CC	13 (19.1)	9 (16.3)	1.20 (0.44–3.28)	0.717
	N = 52 (%)	N = 70 (%)		
Females				
TT	23 (44.2)	24 (34.3)	1	-
TC	14 (26.9)	34 (48.6)	0.43 (0.19–1.01)	0.048
CC	15 (28.9)	12 (17.1)	1.30 (0.50–3.37)	0.583
Age	Cases	Controls		
	N = 48 (%)	N = 50 (%)		
≤60				
TT	12 (25)	8 (16)	1	-
TC	14 (29.2)	24 (48)	0.39 (0.13–1.18)	0.092
CC	22 (45.8)	18 (36)	0.82 (0.27–2.42)	0.713
	Cases	Controls		
	N = 72 (%)	N = 75 (%)		
>60				
TT	16 (22.2)	13 (17.4)	1	1
TC	25 (34.7)	31 (41.3)	0.66 (0.27–1.61)	0.357
CC	31 (43.1)	31 (41.3)	0.81 (0.34–1.97)	0.645

**Table 5 biomedicines-11-02341-t005:** Distribution of *miR-196a2* genotypes in terms of age and gender.

Genotypes	Cases, N = 68 (%)	Controls, N = 55 (%)	OR (95% CI)	*p*-Values
Males				
CC	28 (41.2)	19 (34.5)	1	-
CT	24 (35.3)	25 (45.5)	0.65 (0.29–1.46)	0.298
TT	16 (23.5)	11 (20)	0.99 (0.38–2.56)	0.979
	Cases, N = 52 (%)	Controls, N = 70 (%)		
Females				
CC	11 (21.1)	28 (40)	1	-
CT	25 (48.1)	23 (32.9)	2.77 (1.13–6.79)	0.025
TT	16 (30.8)	19 (27.1)	2.14 (0.82–5.62)	0.118
Age	Cases, N = 47 (%)	Controls, N = 80 (%)		
≤60				
CC	15 (31.9)	27 (33.7)	1	-
CT	21 (44.7)	32 (40)	1.18 (0.51–2.73)	0.697
TT	11 (23.4)	21 (26.3)	0.94 (0.36–2.47)	0.905
	Cases, N = 73 (%)	Controls, N= 45 (%)		
>60				
CC	24 (32.8)	20 (44.4)	1	-
CT	28 (38.4)	16 (35.6)	1.46 (0.62–3.43)	0.386
TT	21 (28.8)	9 (20)	1.94 (0.73–5.19)	0.181

**Table 6 biomedicines-11-02341-t006:** Distribution of *miR-149* and *miR-196a2* genotypes in terms of smoking and alcohol use.

*miR-149* T>C Genotypes	Smokers N = 35 (%)	Non-Smokers N = 77 (%)	OR (95% CI)	*p*-Values
TT	14 (40)	36 (46.8)	1	-
TC	11 (31.4)	24 (31.2)	1.18 (0.46–3.03)	0.733
CC	10 (28.6)	17 (22)	1.51 (0.56–4.10)	0.414
	Alcohol drinkers N = 32 (%)	Non-drinkers N = 80 (%)		
TT	18 (56.3)	32 (40)	1	-
TC	8 (25)	27 (33.8)	0.53 (0.19–1.40)	0.196
CC	6 (18.7)	21 (26.2)	0.51 (0.17–1.48)	0.213
*miR-196a2* C>T Genotypes	Smokers N = 35 (%)	Non-smokers N = 77 (%)		
CC	15 (42.8)	23 (29.8)	1	-
CT	10 (28.6)	33 (42.9)	0.47 (0.18–1.22)	0.115
TT	10 (28.6)	21 (27.3)	0.73 (0.27–1.98)	0.535
	Alcohol drinkers N = 32 (%)	Non-drinkers N = 80 (%)		
CC	14 (43.8)	24 (30)	1	-
CT	8 (25)	35 (43.8)	0.39 (0.14–1.08)	0.066
TT	10 (31.3)	21 (26.3)	0.82 (0.30–2.22)	0.691

**Table 7 biomedicines-11-02341-t007:** Distribution of *miR-149* T>C genotypes in tumor stages and grades.

	TT	TC	CC	*p* Values
	N (%)	N (%)	N (%)	
Tumor grade				
G1	7 (50)	4 (28.6)	3 (21.4)	0.998
G2	29 (42.6)	23 (33.8)	16 (23.6)	
G3	15 (45.5)	10 (30.3)	8 (24.2)	
G4	2 (40)	2 (40)	1 (20)	
Tumor stage				
T1	4 (28.6)	4 (28.6)	6 (42.8)	0.179
T2	11 (57.9)	5 (26.3)	3 (15.8)	
T3	32 (42.1)	29 (38.2)	15 (19.7)	
T4	6 (54.5)	1 (9.1)	4 (36.4)	

**Table 8 biomedicines-11-02341-t008:** Distribution of *miR-196a2 C>T* genotypes in tumor stages and grades.

	CC	CT	TT	*p* Values
	N (%)	N (%)	N (%)	
Tumor grade				
G1	2 (14.2)	9 (64.3)	3 (21.5)	0.149
G2	28 (41.2)	29 (42.7)	11 (16.1)	
G3	12 (36.4)	13 (39.4)	8 (24.2)	
G4	1 (20)	1 (20)	3 (60)	
Tumor stage				
T1	6 (42.8)	4 (28.6)	4 (28.6)	0.034
T2	2 (10.5)	7 (36.8)	10 (52.7)	
T3	36 (47.4)	27 (35.5)	13 (17.1)	
T4	4 (36.3)	3 (27.2)	4 (36.4)	

## Data Availability

All data used for the research are presented in the tables in this article.

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
