# Peer review of "Association of miR-149 T>C and miR-196a2 C>T Polymorphisms with Colorectal Cancer Susceptibility: A Case-Control Study"

_biomedicines, 2023, doi:10.3390/biomedicines11092341_

Round 1

Reviewer 1 Report

 In this manuscript, the relationship between miR-149 T>C (rs2292832) and miR-196a2 C>T (rs11614913) non-coding RNA polymorphisms and the risk of developing CRC in the Azerbaijani population was investigated. The association between CRC and miRNA polymorphisms has been repeatedly described in the literature and correlated to different populations. My main criticism concerns the number of subjects analysed, I think it is too small to reach conclusions that are statistically significant and representative of the entire Azerbaijani population. Therefore, this limit must be taken into account and if it is not possible to expand the case study, at least discuss this in the manuscript.

Author Response

Comment:

The association between CRC and miRNA polymorphisms has been repeatedly described in the literature and correlated to different populations. My main criticism concerns the number of subjects analysed, I think it is too small to reach conclusions that are statistically significant and representative of the entire Azerbaijani population. Therefore, this limit must be taken into account and if it is not possible to expand the case study, at least discuss this in the manuscript.

Answer:

We would like to thank you for the valuable comments. The alterations were made throughout the manuscript (lines 298-311).

Reviewer 2 Report

Biomedicines-2476526

In this case control study titled “Association of miR-149 T>C and miR-196a2 C>T polymorphisms with the colorectal cancer susceptibility: a case-control study” by Bayramov et al., the authors aimed to investigate the association between a genetic variation of miR-149 T>C (rs2292832) and miR-196a2 C>T (rs11614913) and the and the risk of developing CRC in the Azerbaijani population. In this study, majority of the work is focused on single nucleotide polymorphism of miRNAs from healthy and Colorectal cancer patient’s blood. Experimental design to provide enough evidence for the argument is not outstanding, however, this study has statistically excellent number of patients and healthy controls but stands to gain from the following concerns.

It seems authors investigated SNPs in biopsy proven colorectal cancer patients and health individuals, how can this be risk stratifying? Did authors perform any screening on high-risk patients. Ideally authors should have investigated polymorphisms by conducting a screening in high-risk patients to risk stratify based on SNPs in miR-149 and miR-196a2.

Authors claims that Polymorphisms were determined using polymerase chain reaction-restriction fragment length polymorphism (PCR-RFLP).

Why Authors choose RFLP over allele specific PCR.

Authors must provide enough evidence to support their single nucleotide polymorphism data when studying a genetic variant that has not been reported in Colorectal cancer before. For example, random (at least from 5 patients and healthy subjects) sequencing of PCR products to confirm the SNP is strongly recommended. False positive results in agarose gel electrophoresis of PCR products have been evidenced in the past.

I request authors to clearly present the results. For example, gel electrophoresis. Author claims that the presence of an allele-specific band of 308 bp with a control band of 212 bp size was considered positive evidence for each allele. Lane 2: Deletion/Deletion but bands 308 bp bands are not present in both the lanes along with internal controls. It is really difficult to understand the results. Please limit the use of single letter abbreviations.

I recommend authors to describe the data obtained in table 1 and figures 1 and 2 in the results section clearly.

Did authors purified PCR product, what if the restriction enzyme acted on genomic DNA?

English language is good. Minor editing and proof reading is required

Author Response

Comments and Suggestions for Authors

Comment 1. It seems authors investigated SNPs in biopsy proven colorectal cancer patients and health individuals, how can this be risk stratifying? Did authors perform any screening on high-risk patients. Ideally authors should have investigated polymorphisms by conducting a screening in high-risk patients to risk stratify based on SNPs in miR-149 and miR-196a2.

Answer:

We would like to thank you for your valuable comments. The suggested alterations were made throughout the manuscript.

We added a few important details in terms of patients’ stratification. We performed an analysis of the polymorphism distribution and assessed the statistical variances concerning tumor grade and stage (Lines 203-216).

Comment 2. Authors claims that Polymorphisms were determined using polymerase chain reaction-restriction fragment length polymorphism (PCR-RFLP). Why authors chose RFLP over allele specific PCR.

Answer: 

Even though PCR-RFLP is considered time consuming and relatively expensive we opted it over allele specific PCR, because it proved itself as a better technique for most diagnostic purposes, whereas the use of allele specific PCR may be considered with caution. Our main concern was ineffective gene amplification in the past.

Comment 3. Authors must provide enough evidence to support their single nucleotide polymorphism data when studying a genetic variant that has not been reported in Colorectal cancer before. For example, random (at least from 5 patients and healthy subjects) sequencing of PC products to confirm the SP is strongly recommended. False positive results in agarose gel electrophoresis of PC products have been evidenced in the past.

Answer: 

We randomly selected 10% of all the samples for an extra round of genotyping for validation purposes

Comment 4. I request authors to clearly present the results. For example, gel electrophoresis. Author claims that the presence of an allele-specific band of 308 bp with a control band of 212 bp size was considered positive evidence for each allele. Lane 2: Deletion/Deletion but bands 308 bp bands are not present in both the lanes along with internal controls. It is really difficult to understand the results. Please limit the use of single letter abbreviations. I recommend authors to describe the data obtained in table 1 and figures 1 and 2 in the results section clearly.

Answer: 

We have made several improvements to enhance clarity in our paper. Specifically, we have rephrased Table 1-i to improve its clarity. Additionally, we have included the PCR band lengths obtained from RFLP analysis in the text. Furthermore, we have revised Figure 1 and Figure 2 to ensure even greater clarity in our visual representations (Lines 101-129). However, it will be important to mention that our results are different from those in comments. Also there is no deletion/deletion, our study included two transitions. 

Comment 5. Did authors purified PC product, what if the restriction enzyme acted on genomic DNA?

Answer:  

Yes. We used QIAquick PCR Purification Kit (Qiagen, Germany).

Round 2

Reviewer 2 Report

Biomedicines-2476526

Authors mentioned in the rebuttal letter that "We randomly selected 10% of all the samples for an extra round of genotyping for validation purposes" but no data has been provided. I would strongly recommend authors to address reviewer's comments with data but not verbally. If authors have data, please include it in either supplementary or main figures section of the manuscript.

Again, for a comment: Did authors purified PC product, what if the restriction enzyme acted on genomic DNA?, author's response was: We used QIAquick PCR Purification Kit (Qiagen, Germany).

Please include it in the methods section.

Author Response

We thank the reviewer for pointing out this issue. According to the reviewer's suggestion, we have changed the manuscript accordingly and uploaded a file with supplementary data.
